# Dismembered Ophiolite of the Olkhon Composite Terrane (Baikal, Russia): Petrology and Emplacement

**Eugene V. Sklyarov [1,2,]*** , **Angrey V. Lavrenchuk [3,4]** , **Valentine S. Fedorovsky [5]** ,
**Evgenii V. Pushkarev [6]** , **Dina V. Semenova [3,4]** and **Anastasia E. Starikova [3,4]**

1   Institute of the Earth's Crust, Siberian Branch of the Russian Academy of Sciences, 128 Lermontov st.,
    664033 Irkutsk, Russia
2   School of Engineering, Far East Federal University, 8 Sukhanov st., 690091 Vladivostok, Russia
3   V.S. Sobolev Institute of Geology and Mineralogy, Siberian Branch of the Russian Academy of Sciences,
    3 Akad. Koptyuga st., 630090 Novosibirsk, Russia; alavr@igm.nsc.ru (A.V.L.);
    semenovadina@gmail.com (D.V.S.); starikova@igm.nsc.ru (A.E.S.)
4   Department of Geology and Geophysics, Novosibirsk State University, 2 Pirogov st.,
    630090 Novosibirsk, Russia
5   Geological Institute, Russian Academy of Sciences, Pyzhevsky per. 6, 119017 Moscow, Russia;
    valentinfedorovskii@mail.ru
6   A.N. Zavaritsky Institute of Geology and Geochemistry, Uralian Branch of the Russian Academy of Sciences,
    Akad. Vonsovsky st., 620016 Ekaterinburg, Russia; pushkarev.1958@mail.ru
*   Correspondence: skl@crust.irk.ru; Tel.: +7-914-908-9408

**Abstract:** Dismembered ophiolites in the Early Paleozoic Olkhon terrane, a part of the Baikal collisional belt in the southern periphery of the Siberian craton, occur as fault-bounded blocks of ultramafic and mafic rocks from a few meters to hundreds of meters in size. The ultramafic rocks are mainly dunite–harzburgite peridotites with gradual transitions between the lithologies, as well as moderate amounts of enstatitite, wehrlite, and clinopyroxenite, but no lherzolite. Most peridotites have strongly depleted chemistry and a mineralogy corresponding to the harzburgite type usual for ophiolites of suprasubduction zones (SSZ). The mafic rocks are leuco- to melanocratic gabbros with different relative percentages of clinopyroxene, olivine, and plagioclase, which enclose thin layers and lenses of clinopyroxenite and anorthosite. They bear back-arc basin geochemical signatures, a setting inferred for the Neoproterozoic southern Siberian craton. The gabbroic rocks are of two geochemical groups; most of their trace-element patterns show Ta-Nb minimums and Sr maximums common to suprasubduction zone ophiolites. Judging by the Ol + Opx + Chl + Chr mineral assemblages, the Olkhon peridotites underwent low amphibolite and amphibolite regional metamorphism at 500–650 °C. The occurrence of the ultramafic and mafic bodies is consistent with formation in an accretionary wedge metamorphosed during a collisional orogeny. The mantle and crustal parts of the Olkhon ophiolite suite apparently were incorporated into the terrane during the frontal collision of perio-oceanic structures with the Siberian craton. Then, in a later oblique collision event, they became dismembered by strike-slip faulting into relatively small bodies and fault blocks exposed in the present erosional surface.

**Keywords:** ophiolite; ultramafic rocks; gabbro; amphibolite; petrology; CAOB; Olkhon terrane; collisional orogeny

## 1. Introduction

Ophiolites represent fragments of upper mantle and oceanic crust that became incorporated into continental margins during accretionary, subduction, obduction, and collision processes ([1–4], etc.).

Recent data on ophiolites from orogenic belts of different ages ([3,4] and references therein) have extended the classical concept [1] of their simplicity and mid-ocean ridge affinity: they turn out to be complexly structured and involve constituents formed in various tectonic settings [4]. Ophiolites are often used as guides to reconstruct the history of orogens of different sizes and ages. They may be well preserved and complete, such as the reference ophiolites of Oman or Troodos [5–12], or others [13,14], but are rather found in many collisional and subduction-accretionary orogens as heavily deformed and dismembered fragments. In this case, identifying the origin of mafic and ultramafic bodies becomes challenging, and the ophiolite suites have to be reconstructed from dispersed pieces. It is especially problematic in high-grade terranes where tectonic and metamorphic effects have obscured the primary igneous structures and mineralogy. Olkhon is such a terrane in the northeastern margin of the Central Asian Orogenic Belt (CAOB). The recent-most overview [15] does not mention ophiolites in this CAOB part, possibly because they are dismembered, though the Baikal-Muya ophiolite belt was described long ago (e.g., [16]), as well as other quite numerous ophiolite remnants.

The Olkhon composite terrane produced by Early Paleozoic collisional orogeny ([17–19] and references therein) is among the best-documented structures of this kind due to its accessibility, good exposure, and location in the scenic landscapes of the Lake Baikal coast. The terrane consists of sedimentary and igneous complexes which differ in age and in the origin of protoliths [19] metamorphosed to amphibolite and granulite facies [20]. Numerous small ultramafic bodies found within the terrane, mainly dunite and harzburgite [21,22], are often interpreted as residual ophiolitic peridotites [23,24], while the presence of other possible members of an ophiolite suite remains questionable. The terrane also accommodates numerous compositionally diverse gabbros that were metamorphosed during collisional orogeny. Some originated in an island arc setting [25–27], but the origin of others is unclear. Deciphering the history of numerous amphibolite bodies, which have lost their primary magmatic structures and textures, as well as primary mineral assemblages, is likewise a challenge. The key point is whether the metamorphic peridotides and gabbros tectonically dispersed within the Olkhon terrane belong to a single ophiolite. This paper focuses on the local geology and compositions of the Olkhon ultramafic and mafic rocks, with implications for the primary ophiolite structure. The available and new data on the tectonic settings of ophiolite and gabbro bodies are reviewed with criticism in the context of ophiolites involved into the regional orogenic framework during collisional processes.

## 2. Geological Background

The Olkhon composite terrane is a part of an Early Paleozoic collisional belt bordering the Siberian craton in the south (Figure 1). It is a collage of subterranes, micro-terranes, and tectonic slices that formed during the Early Paleozoic frontal and oblique collisions of Paleo-Asian oceanic structures with the craton [17,18]. The rocks of these tectonic units differ in the compositions and ages of their protoliths [19]. The largest and most homogeneous Krestovsky subterrane in the southwestern Olkhon area (Figure 2) comprises a 500 Ma volcanoplutonic complex of metavolcanics with subduction geochemical affinity and large gabbro intrusions, such as the Birkhin complex [25–27], as well as less abundant 470 Ma subalkaline gabbro of the Ust'-Krestovsky complex and 460–470 Ma syenites, including nepheline varieties, and granites [26,28,29].

The northeastern part of the Olkhon terrane consists of gneisses and gneissic granites predominant over amphibolites, calcitic and dolomitic marbles, quartzite, and calcareous-siliceous rocks [18,19]; numerous small granite and pegmatite intrusions and thousands of veins; several relatively large gabbro massifs; and several small ultramafic bodies [21,22].

The rocks of the terrane underwent metamorphism of granulite (500 Ma) and low amphibolite (450–470 Ma) facies [20] concurrent with collisional deformations. The combined processes produced heavily faulted elongated rock complexes which are hard to interpret in terms of tectonic setting and age. Recent high-resolution mapping based on satellite imagery and field data [30–34] has undermined the previous model that predicted the presence of weakly deformed volcanic–sedimentary sequences

in the terrane [35]. The apparent stratigraphic sequence of sedimentary and igneous rocks is actually a series of tectonic slices resulting from large-scale thrust and strike-slip faulting.

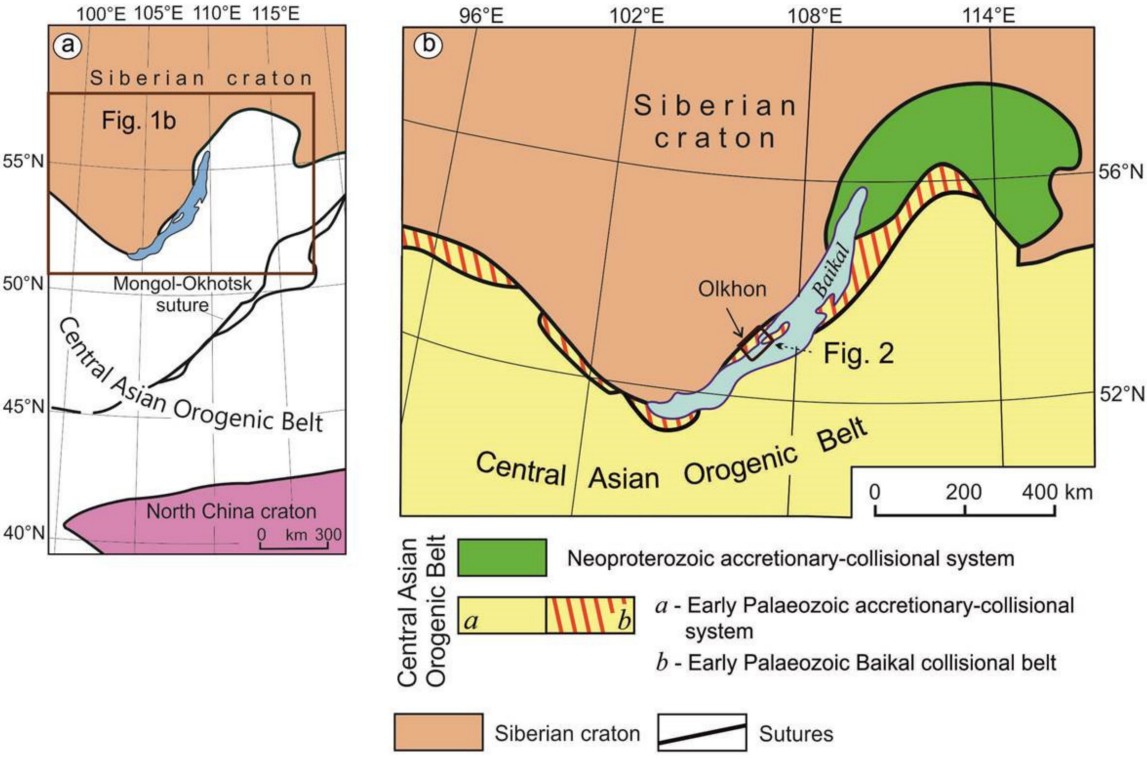

**Figure 1.** Simplified tectonics of Central Asia (**a**) and terranes in the Early Paleozoic Baikal collisional belt of northern Central Asian Orogenic Belt (CAOB) (**b**), modified after [19].

This is valid also for the Olkhon ophiolite. Although the ultramafic rocks were identified as ophiolite-type residual peridotite without much doubt [23], the presence of ophiolitic gabbros, sheeted dike complexes, and volcanics has been overlooked. The age and geodynamic constraints are available for only two complexes of the Krestovsky subterrane, out of countless gabbro bodies and blocks from a few tens of meters to a few km in size: the 500 Ma Birkhin gabbro with distinct suprasubduction fingerprints [25–27] and the 470 Ma Ust-Krestovsky complex of subalkaline gabbro which presumably resulted from interaction of a mantle plume with suprasubduction lithospheric mantle [26]. The Ust'-Krestovsky gabbro bodies occur as a small intrusion and individual dikes, including those combined with granosyenite and syenite [26,36,37]. Other gabbro occurrences in the Olkhon terrane remain poorly investigated, and their tectonic settings are still unclear. The widespread amphibolites, which were traditionally interpreted as constituents of volcanic-sedimentary sequences on the basis of geochemistry [38], are poorly understood as well. The mafic rocks have lost their primary magmatic features during tectonic and metamorphic events, and only ultramafics are reliably detectable as fragments of an ophiolite suite.

## 3. Ultramafic and Mafic Rocks in the Olkhon Terrane

Round or lens-shaped bodies of peridotites are widespread throughout the northeastern Olkhon area (Figure 2) in chains aligned with metamorphic rocks. Their sizes are most often within a few tens of meters, except for the Tog and Kharikta peridotites that exceed 500 m. Their contacts with the gneiss, amphibolite, and marble hosts are sharp and free from signatures of metasomatic interaction, structural transformations, and replacement of primary mineral assemblages. There are also fine-grained garnet–pyroxene–plagioclase rocks or garnet amphibolites that occur only in association with the peridotites. The latter are dunite or harzburgite, with a minor amount of wehrlite, clinopyroxenite,

and spinel–olivine–enstatite rocks. Several bodies are quite large and well exposed, including those described previously [23,24,39].

Shida (1 in Figure 2): five peridotite lens-shaped and round bodies of different sizes [23] occurring among strongly deformed marbles, amphibolites, migmatized garnet-biotite gneisses, and blastomylonites (Figure 3). Peridotites are dunite and harzburgite intruded by numerous dikes and veins of plagiogranite and plagioclaiste.

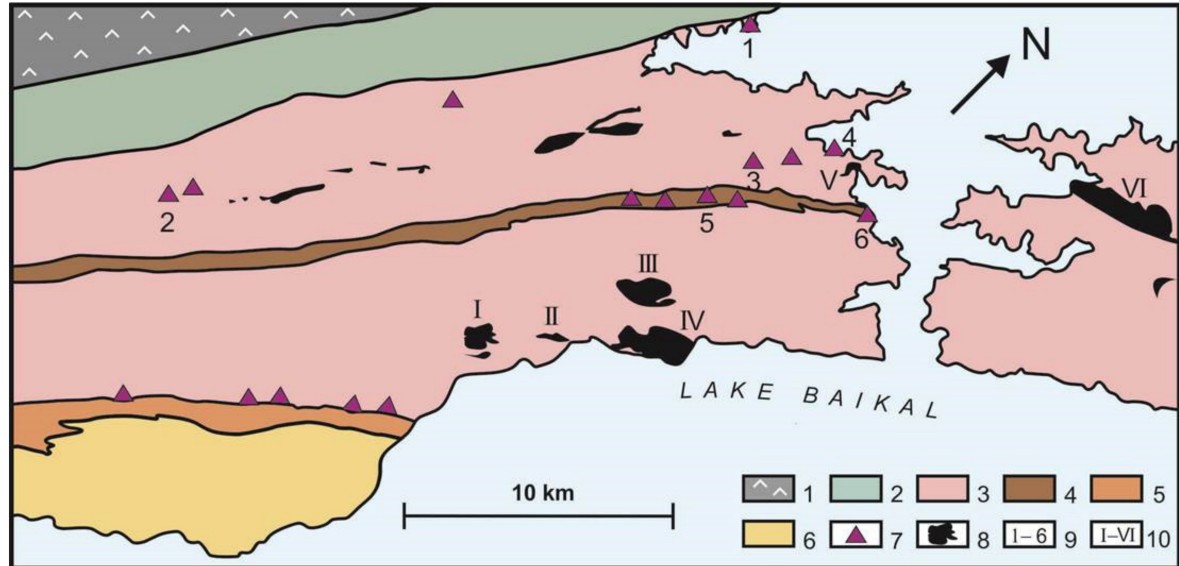

**Figure 2.** Ophiolitic ultramafics and gabbro in the northeastern Olkhon area. 1 = Paleoproterozoic igneous and metamorphic complexes of the Siberian craton. 2–8 = igneous and metamorphic complexes of the Olkhon terrane: marbles, gneisses, and mafic granulites (2); a complex of predominant gneissic granites and less abundant amphibolites, marbles, and quartzites (3); main strike-slip zone composed of amphibolites (4); Orso microterrane between the Krestovsky island arc subterrane and a collage of microterranes (5); island arc carbonate-amphibolite rocks of the Krestovsky subterrane with large gabbro bodies (6); small peridotite bodies (7); gabbro (8). Arabic numerals stand for names of peridotite bodies: Shida (1), Turpan (2); Bezymyanny (3); Bayar (4); Tog (5); Sakhyurta (6). Roman numerals stand for names of gabbro bodies: Orgoita (I); Treugolnik (II), Tankhan (III); Gantel (IV); Krest (V); Olkhon (VI).

Turpan (2 in Figure 2): two closely spaced harzburgite boudines (the larger one is >100 m long) and several orthopyroxenite bodies among biotite- or garnet–biotite gneisses and pegmatitic granites (Figure 4), a part of a longer NNE chain of peridotitic lenses. There are a few small outcrops of fine-grained hercynite–enstatite and hercynite–olivine–enstatite rocks among harzburgite in the southern part of the ultramafic body, but their relationships with harzburgite host are unclear. The peridotites are associated with amphibolites, often migmatized, which contain garnet if occurring in the vicinity of the ultramafic rocks. The peridotites are locally injected by carbonate veins.

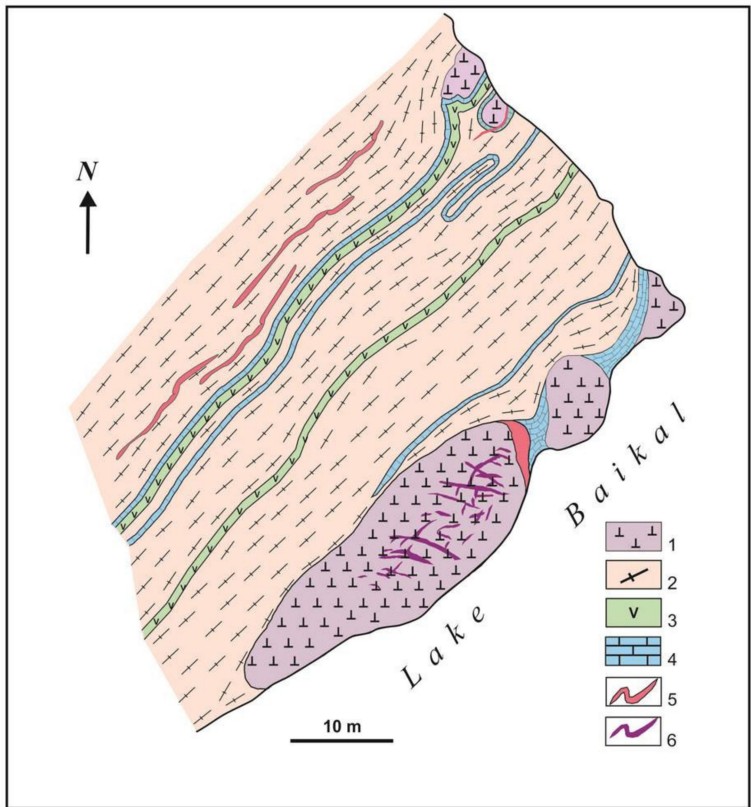

**Figure 3.** Geological map of Shida Cape, modified after [23]. 1 = dunite; 2 = migmatized biotite–garnet gneiss and blastomylonite; 3 = amphibolite; 4 = calcitic marble; 5 = granite veins; 6 = plagioclasite veins.

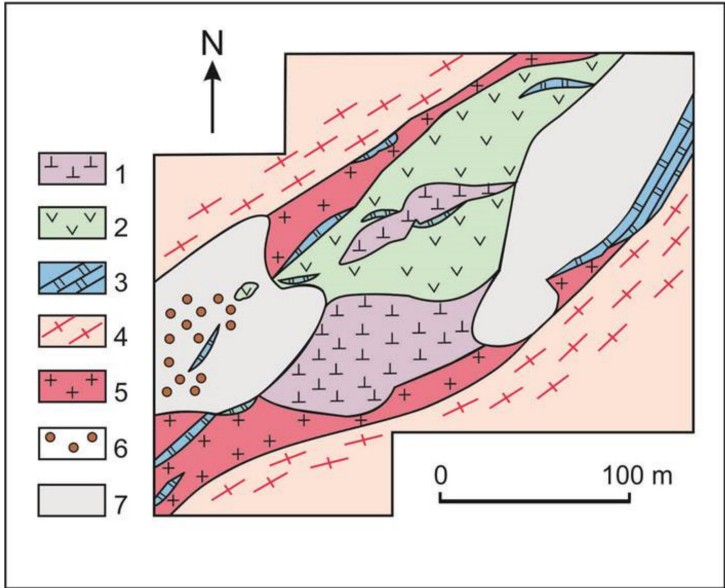

**Figure 4.** Geological map of Turpan peridotite. 1 = harzburgite and enstatitite; 2 = amphibolite, garnet amphibolite; 3 = calcite-dolomite marble; 4 = gneisss and gneissic granite; 5 = gneissic granite and pegmatite; 6 = scattered geyserite; 7 = quaternary deposits.

Bezymyanny (3 in Figure 2): a block one in a chain of ultramafic bodies among amphibolites, within a few tens of meters; its geological mapping is problematic because of poor exposure.

Bayar (4 in Figure 2): a 120 × 70 m body in the shore of Lake Baikal (Figure 5), composed of dunite and harzburgite grading smoothly one to another and occurring among garnet and

garnet–anorthite–fassaite amphibolites; the peridotite and metamorphic rocks have a sharp contact without signatures of interaction.

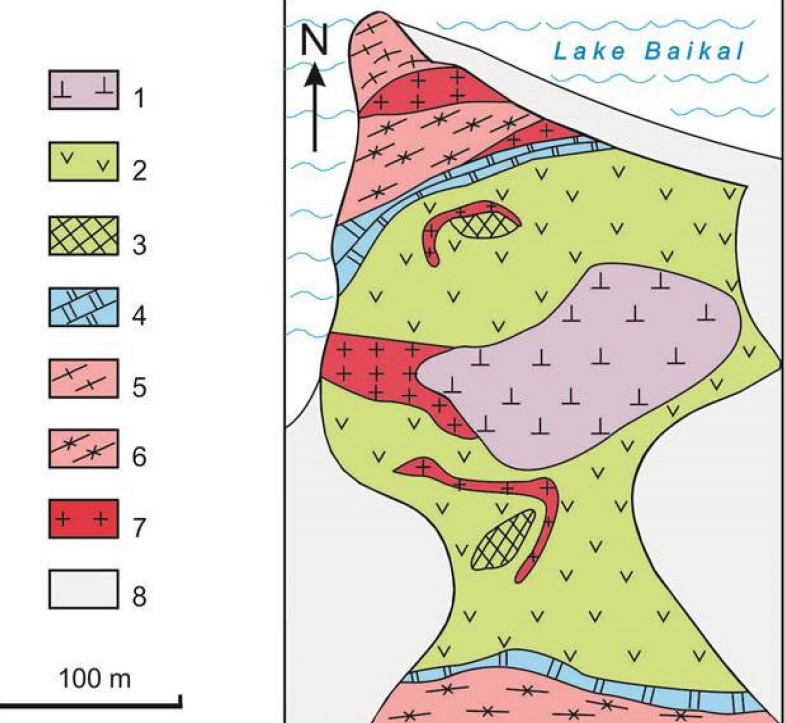

**Figure 5.** Geological map of Bayar peridotite. 1 = dunite and harzburgite; 2 = amphibolite; 3 = garnet amphibolite and garnet–anorthite–fassaite rocks; 4 = calcite–dolomite marble; 5 = biotite and garnet–biotite gneisses; 6 = gneissic granite; 7 = pegmatite; 8 = quaternary deposits.

Tog (5 in Figure 2): one of the largest bodies, with a drop-like shape (Figure 6), located in the Main Shear Zone, which accommodates a chain of ultramafic bodies, including Sakhyurta (see below). The body mainly consists of harzburgite with irregular dunite patches grading smoothly one to another. It occurs among amphibolites and has a sharp contact with metamorphic rocks without signatures of interaction. This is a unique peridotite structure in the Olkhon terrane, which encloses exotic mafic and ultramafic fragments. The mafic fragments, from a few meters to a few tens of meters, are foliated and partly migmatized amphibolites with gneissic structures, including garnet varieties. They form a chain almost aligned with the peridotitic intrusion and may be remnants of a dolerite dike that was boudinated and deformed during synmetamorphic deformation. The ultramafic fragments, from tens of centimeters to a few meters, have a mineralogy consisting of clinochlore and amphibole (in different proportions, from amphibole-free chlorite–magnetite to amphibole varieties), as well as magnetite, rarely pyroxene, and occasionally minor percentages of Cr-free spinel. The amphibole–magnetite–chlorite inclusions in peridotites are randomly distributed and often bear significant amounts of zircon.

Sakhyurta (6 in Figure 2): several ultramafic bodies exposed in a cliff (Figure 7), among calcite and calcite–dolomite marbles, biotite, and garnet–biotite gneisses and amphibolites. All ultramafic bodies are mostly dunitic with phlogopite–plagioclase veins and occasionally with small enstatite veins. The contact with the host rocks is sharp. The amount of calcite in dunite decreases from 30% immediately at the contact (Figure 8a,b) to a minor percentage 3 m away. The relations of calcite with silicate minerals show that the mineral assemblage is equilibrated. Granite veins crosscut both the peridotites and their hosts [39].

Another chain of peridotites is located in the south, where a collage of tectonic plates borders the Orso microterrane (Figure 2). Unlike the other bodies, which have preserved their textural features

and mineral assemblages, the small ultramafic lenses here have been completely transformed to talc–chlorite–tremolite schists compositionally similar to pyroxenite.

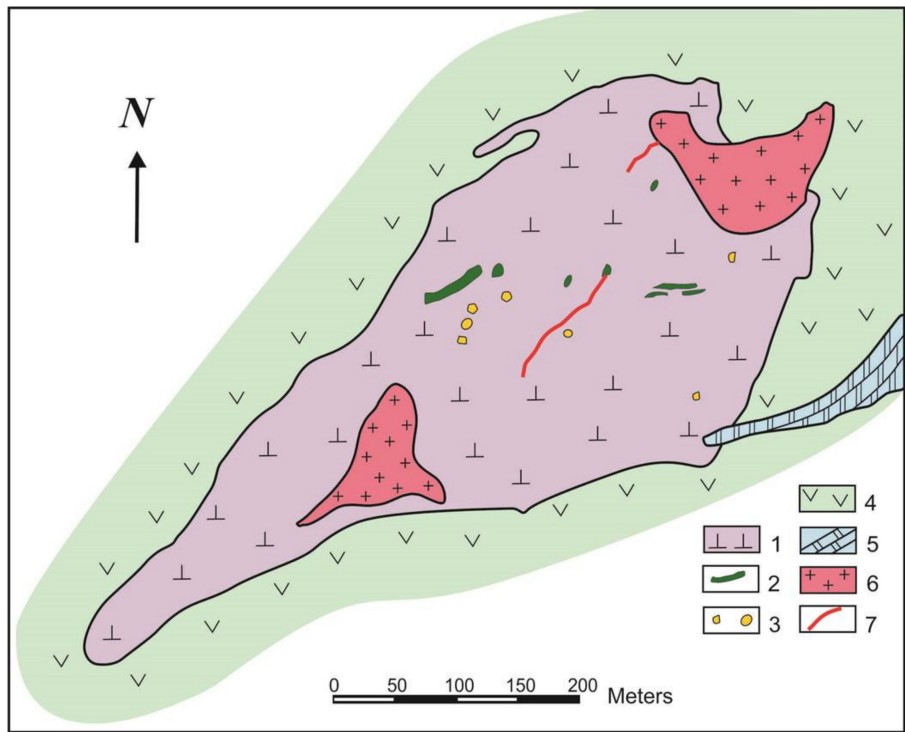

**Figure 6.** Geological map of Tog peridotite. 1 = dunite and harzburgite; 2 = amphibolite and garnet amphibolite in peridotites; 3 = magnetite–amphibole–chlorite rocks in peridotites; 4 = amphibolite; 5 = calcite marble; 6 = granite and pegmatite; 7 = pegmatite veins.

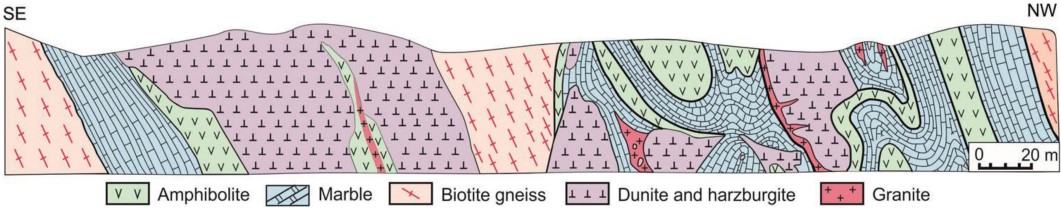

**Figure 7.** Geological cross section of Sakhyurta peridotite in a cliff exposure, modified after [39].

Gabbro bodies are larger than the peridotites and occasionally exceed 1 km (Figure 2). The largest ones are Orgoita (I in Figure 2), Treugolnik (II in Figure 2), Tankhan (III in Figure 2), Gantel (IV in Figure 2), Krest (V in Figure 2), and Olkhon (VI in Figure 2). Some are represented by differentiated sequences from olivine gabbro to leucogabbro with less abundant olivine clinopyroxenite, clinopyroxenite, and anorthosite. All contacts of gabbro with the host carbonates are tectonic-related with rolling during synmetamorphic deformations. Some of the gabbro underwent amphibolite facies metamorphism and transformed to amphibolite. The intrusion margins are often scarned with the formation of crosscutting garnet–pyroxene veins.

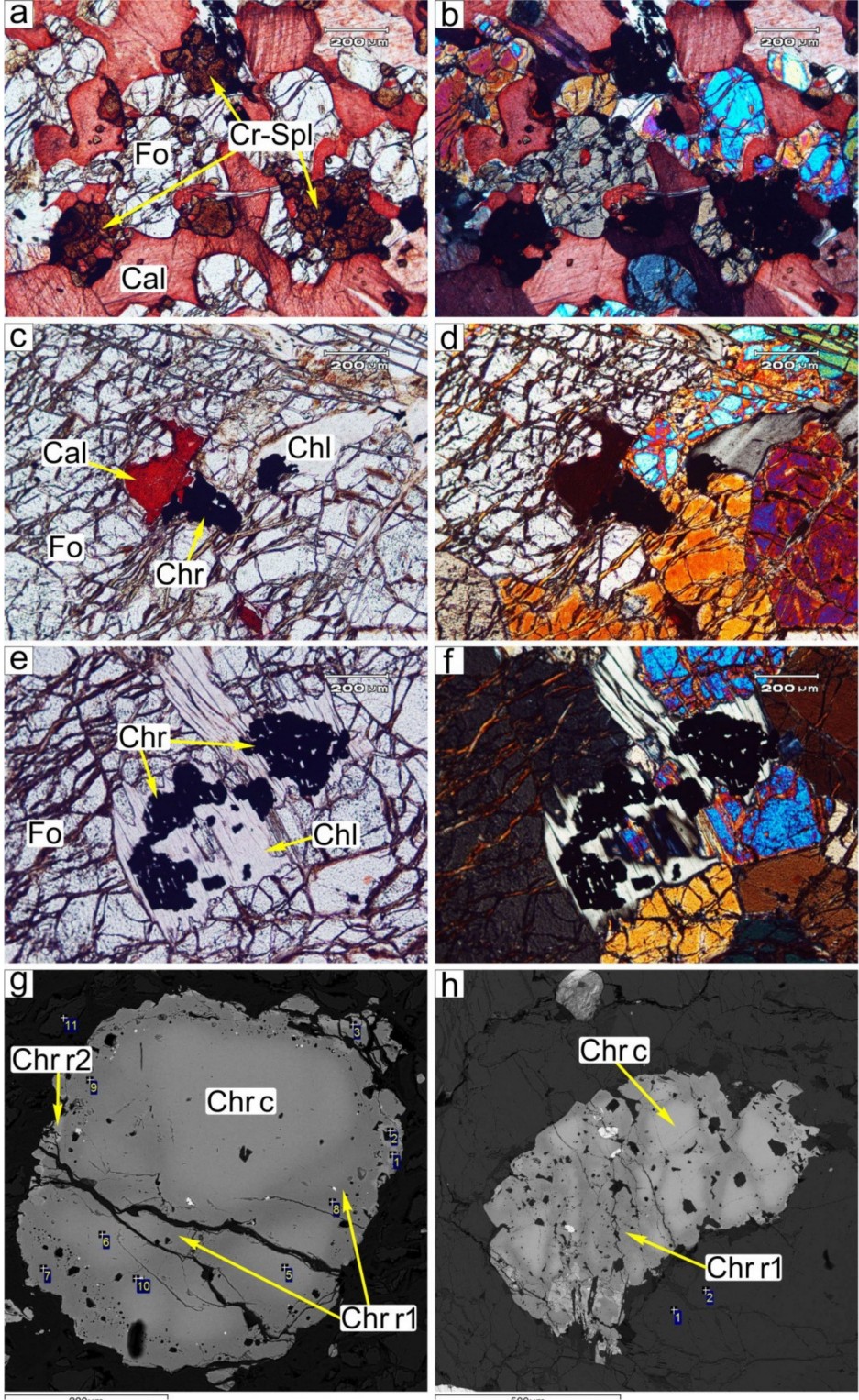

**Figure 8.** Microphotographs (**a**–**f**) and back-scattered electron (BSE) images (**g,h**) of ultramafic rocks. (**a,b**): calcite–forsterite rocks at the contact of the Sakhyurta peridotite with marble (cross-polarized light); (**c,d**): calcite-bearing dunite at 3 m from the contact with marble, Sakhyurta peridotite; (**e,f**): chrome spinel aggregate in chlorite (Tog dunite); (**g,h**): zoned Cr spinel in chlorite (Tog dunite).

Amphibolites are widespread within the terrane, but their interpretation as ophiolite metavolcanics or metagabbro is questionable. The ophiolitic nature of the exposed amphibolites can hardly be

explained in terms of the model [35,39] considering them as part of volcanic–sedimentary sequences with abundant graywacke and carbonates. Their origin can be reinterpreted in a new tectonic model [17,18] which explains the structure of the Olkhon terrane as a collage of thrust sheets that originated in different settings and became tectonically juxtaposed along shear zones. According to this model, the amphibolites which have no stratigraphic or geological relations with gneisses and marbles do not belong to metagraywacke and metacarbonate sequences but rather represent tectonic sheets. Some amphibolites may be synmetamorphic dikes [36,37], but this interpretation is applicable to the least metamorphosed and deformed bodies. Currently, it is impossible to discriminate unambiguously between synmetamorphic dikes or larger intrusions and possible ophiolitic volcanic and subvolcanic rocks.

Thus, the Olkhon ophiolite complex is inferred to include peridotites, gabbro, and metagabbro and probably some amphibolites derived from volcanic and subvolcanic rocks. The ophiolite affinity of peridotite and gabbro is perfectly supported by their mineralogy and major- and trace-element compositions (see below), but the origin of amphibolites is not always clear and open to discussions.

## 4. Materials and Methods

Representative samples of ophiolitic rocks were analyzed for major, trace, and rare-earth elements. Major elements were determined by X-ray fluorescence spectrometry (XRF) at the Center for Geodynamics and Geochronology of the Institute of the Earth's Crust (Irkutsk). Trace elements and rare earths were determined by inductively coupled plasma mass spectrometry (ICP–MS) on an Agilent Technologies Agilent 7500ce analyzer (USA) at the Limnological Institute (Irkutsk), at the Analytical Center Ultramikroanaliz (analyst S. Panteeva). For ICP–MS, the samples were fused with $LiBO_2$ following the standard procedure. Calibrations were with internal and international standards G-2, GSP-2, JG-2, and RGM-1. Analytical accuracy was within 5% for trace elements and rare earth elements (REE).

Minerals were analyzed at the Center for Multi-Elemental and Isotope Research of the V.S. Sobolev Institute of Geology and Mineralogy (Novosibirsk) by scanning electron microscopy with energy-dispersive X-ray spectroscopy (SEM–EDS) on a Tescan MIRA 3 LMU scanning electron microscope with an Oxford Inca Energy 450+/Aztec Energy XMax-80 and Inca Wave 500 microanalyzers. The operation conditions for EDS were: 20 keV beam energy, 1.5 nA beam current, and 20 s spectrum live acquisition time. The results were checked against synthetic compounds and natural minerals: $SiO_2$ (O, Si), $BaF_2$ (F, Ba), $NaAlSi_3O_8$ (Na), $MgCaSi_2O_6$ (Mg, Ca), $Al_2O_3$ (Al), and $Ca_2P_2O_7$ (P). Matrix correction was performed with the XPP algorithm as part of the built-in Inca Energy software.

The U-Pb ages of zircons were measured by the LA–SF–ICP–MS method on a Thermo Fisher Scientific Element XR magnetic sector-field ICP-MS coupled to a New Wave Research UP-213 Nd:YAG laser at the Analytical Center for Multi-Element and Isotope Research SB RAS of the V.S. Sobolev Institute of Geology and Mineralogy (Novosibirsk). The ICP–MS was optimized using continuous ablation of a NIST SRM 612 reference glass to provide maximum sensitivity while maintaining low oxide formation 248ThO+/232Th+ratios (<2%). All measurements were done using electrostatic scanning (E-scan) at masses 202Hg, 204(Pb + Hg), 206Pb, 207Pb, 208Pb, 232Th, and 238U. The signals were detected in the counting mode for all isotopes except for 238U and 232Th, for which the triple mode was applied. The zircons were analyzed by a laser of 80-μm diameter with fluence of 1.5–2.0 $J/cm^2$ and repetition rate of 5 Hz. Data reduction was carried out with the software package Glitter [40]. Data were corrected for U-Pb fractionation during laser ablation and for instrumental mass discrimination by standard bracketing with repeated measurements of the Plešovice zircon reference material [41]. GJ-1 zircon reference material [42] (Jackson et al., 2004) was used as an unknown for independently controlling the reproducibility and accuracy of the corrections. Concordia ages and diagrams were generated using the *Isoplot 3.00* software package (other information is in Reference [43]). The errors are quoted at the one sigma level (%) for individual analyses and as two sigma absolute values for the concordant ages. The correction for common lead was following [44].

## 5. Petrography and Mineralogy of the Olkhon Ultramafic and Mafic Rocks

Peridotites in the Olkhon composite terrane are mostly dunite and harzburgite accompanied by smaller bodies of orthopyroxenite and rare clinopyroxenite. They consist of medium- or coarse-grained olivine, enstatite, Cr-spinel, and diopside (in clinopyroxenites) and minor amounts of clinochlore, but no serpentine. Some samples bear carbonate minerals (calcite) or less often dolomite or magnesite. Dunites mainly have a granoblastic texture, while the textures in harzburgites are most often porphyro- or poikiloblastic, with olivines either forming a fine-grained groundmass or occurring as idioblasts in relatively coarse orthopyroxenes. Chromian spinel sometimes occurs as small inclusions in olivine but more commonly as grain clusters or intergrowths in chlorite (Figure 8e,f). Chlorite is interstitial rather than replacing olivine or enstatite, which, along with the presence of chromite, is evidence of their concurrent crystallization and equilibrium of the chlorite assemblage with olivine, enstatite, and Cr-spinel. Pentlandite is a common opaque phase.

Mineral chemistry data (Tables S1 and S2 in Supplementary Materials) show notable variations in the rock-forming olivine and orthopyroxene and in accessory chromian spinel from the peridotites ([23] and our analyses). The Mg# values (Mg/(Mg + Fe$_{tot}$) atomic ratio) in olivine and orthopyroxene vary from 0.94 to 0.87. The variations in accessory spinel (Table S1 in Supplementary Materials; Figure 9) are even larger: its Cr/(Cr + Al) ratios are from 0 to 1 and form a statistically prominent cluster at Cr# = 0.5–0.9 and Mg# of 0.2 to 0.6 (Figure 10). Most of the data points in its Cr#–Mg# diagram (Figure 10) plot in the top right of the fore-arc peridotite field, but many fall outside this field, indicating possible metamorphic origin [45].

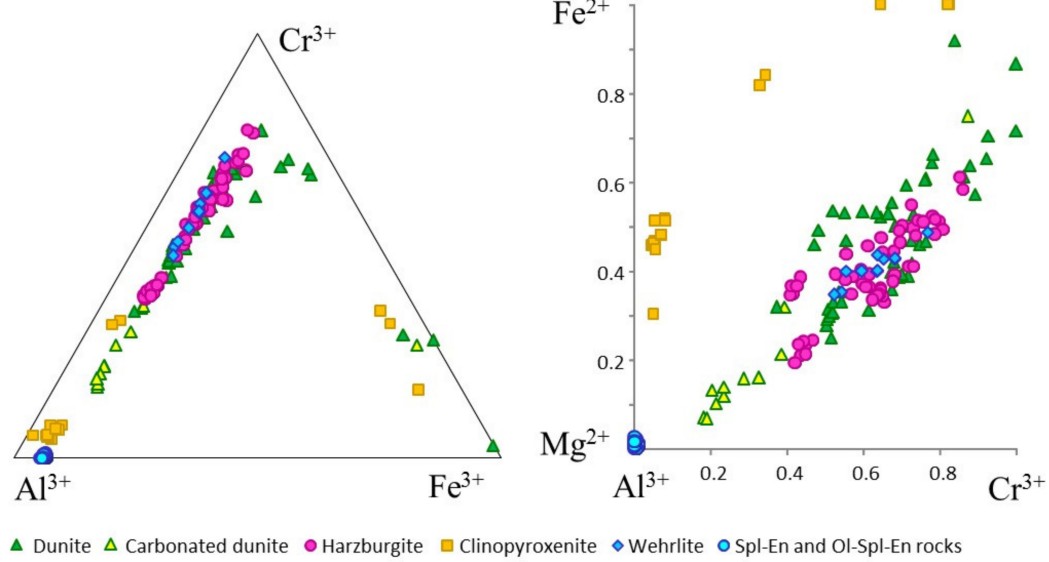

**Figure 9.** Chromite and spinel from the Olkhon peridotites.

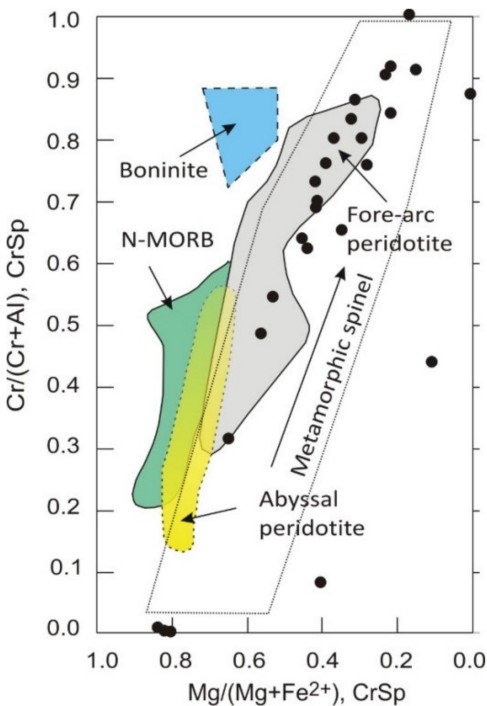

**Figure 10.** Data for spinel from the Tog and Shida peridotites. Abyssal peridotites, boninite, N-MORB are from [46,47]. Data for spinel in fore-arc peridotites are from [48,49]. The field of metamorphic spinel is after [45].

The latter inference is further supported by correlation between Cr# in chromian spinel and forsterite percentage in olivine (Figure 11). Only a part of their Cr#–Mg# points fall within the olivine–spinel mantle array (OSMA) common to mantle peridotite, but about a half of the compositions rather plot within the metamorphic field (Figure 11). Some of the spinels are zoned (Figure 8g,h), with relict chromite in the cores having the OSMA values of Cr/(Cr + Al) = 0.65–0.8 and Mg# ~0.3. The cores are replaced and overgrown with Al- and Mg-rich spinel (Table S1 in Supplementary Materials) along grain defects (cracks, etc.) of certain orientations (see the back-scattered electron (BSE) images in Figure 8g,h). Therefore, the chromian spinel underwent alteration long after its origin in the upper mantle and still later was re-equilibrated and recrystallized during amphibolite and low amphibolite metamorphism within the stability field of magnesian chlorite found intergrown with spinel in interstices of the peridotites (Figure 8e–h). This appears to be the case for chromite with >55% $Cr_2O_3$ and high FeO but low MgO and $Al_2O_3$ (<6%) had formed. This spinel may have a high percentage of the chromite end-member component and $Al_2O_3$ as low as 1%–3%, indicating its equilibrium with chlorite [45,50,51]. The chromite forms rims of zoned grains (Figure 8g,h) and tends to the Cr apex in the diagram of trivalent cations (Figure 9). It plots along the 500 °C solvus isotherms [52]; i.e., its late high-Cr phase formed at temperatures no higher than 600 °C, most likely within the 400–550 °C range. Chromite of this kind may coexist with high-forsterite olivine ($Fa_{4-6}$) in the Shida peridotite [23]. Chromium-bearing magnetite (Table S2 in Supplementary Materials) formed in oxidized conditions when peridotites ascended to the upper crust. Unlike other peridotites, that of Sakhyurta contains up to 30% calcium carbonate at the contact with marble and chromium-bearing hercynite with #Cr = 0.25–0.47 (Figure 9).

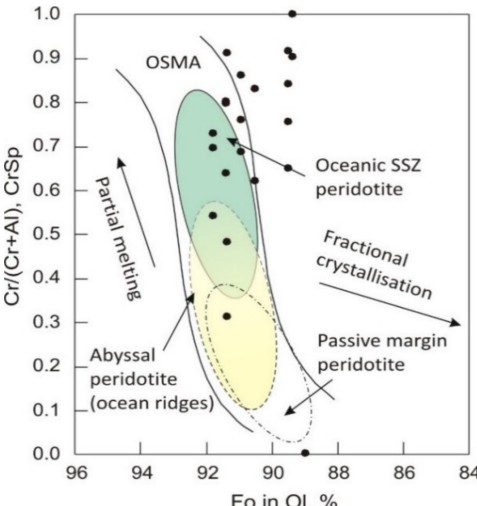

**Figure 11.** Compositional relationships between Fo content of olivine and Cr# of Cr spinel from the Tog and Shida peridotites. The olivine = spinel mantle array (OSMA) is after [53,54].

Gabbros have leucocratic or melanocatic compositions, according to relative percentages of rock-forming clinopyroxene, olivine, and plagioclase, and enclose thin clinopyroxenite and anorthosite layers and lenses. Olivine and clinopyroxene are early phases; plagioclase is an intercumulus phase in clinopyroxenite but is part of a cumulus assemblage in ophytic gabbro and leucogabbro. The Tankhan body is composed of pyroxenite–melagabbro–gabbro–leucogabbro–anorthosite rhythms. The Gantel gabbro and leucogabbro enclose xenoliths of pyroxenite which may have formed earlier in intermediate magma chambers. Elongated plagioclase crystals in some gabbro samples show thrachytic-like textures with preferred orientations related to magma flow. Another important petrographic feature in gabbro is the presence of abundant late magmatic brown amphibole in interstices between early cumulus minerals; the amphibole grains occur as poikilitic crystals with numerous inclusions of subhedral plagioclase and clinopyroxene. The rock textures bear signatures of postmagmatic tectonic effects: ductile bending and brittle fracture in plagioclase twins. Olivine gabbro shows zoned coronite textures with Ol–OPx–Amph–Amph+Sp–Pl zones at the olivine–plagioclase contact. Such coronitic textures are known from many gabbros worldwide [55–58] and may result either from amphibolite or granulite facies metamorphism or from subsolidus Ol–Pl interaction in the presence of a fluid.

Some gabbros experienced postmagmatic alteration due to injections of granite dikes. Olivine and clinopyroxene are replaced by green amphibole though plagioclase that was only slightly saussuritized. The effect of later dikes on some gabbros produced metasomatic skarn associations with abundant green spinel, carbonate, and coarse crystals of green amphibole (3–5 cm across).

Garnet–pyroxene–anorthite rocks and garnet amphibolite either form small bodies adjacent or proximal to peridotites (Bayar, Turpan) or occur as small fragments within the Tog peridotite. They have fine- or medium-grained textures; the structures are sometimes massive but more often gneiss-like in strongly migmatized varieties. The modal mineral composition includes pyroxene, garnet, plagioclase, and quite often amphibole. Mafic minerals show large ranges of compositions. Pyroxenes commonly have low-Na diopside–hedenbergite compositions, but some are fassaitic and contain up to 15 wt.% $Al_2O_3$ (Table S3 in Supplementary Materials). Garnets are likewise highly variable in composition (Table S4 in Supplementary Materials) from grossular-pyrope varieties up to grossular-rich almandine (Ca-comp. = 40% to 75%) with only <10% andradite. Such garnet compositions are common to HP metamorphic mafic rocks. Plagioclases are, on the contrary, homogeneous and approach pure anorthite ($X_{An}$ = 0.95–1.0).

The Turpan pleonast–enstatite and olivine–pleonast–enstatite rocks typically lack chromite but contain high-Mg (Mg# = 0.82–0.83) and low-Cr (0.26–0.72 wt% $Cr_2O_3$) spinel (Table S2 in Supplementary Materials; Figure 9). Olivine in these rocks is compositionally similar to that in the peridotite

(Table S1 in Supplementary Materials), while enstatite has higher contents of $Al_2O_3$ (1.8–2.6 wt%). The accessory phases commonly include Mg-ilmenite with 7.7 to 9.3 wt% MgO.

## 6. Major- and Trace-Element Compositions of the Olkhon Ultramafic and Mafic Rocks

Peridotites vary largely in $MgO/SiO_2$ ratios, in accordance with the relative percentages of olivine and orthopyroxene in the dunite–harzburgite series (Table S5 in Supplementary Materials). They plot a single trend in the $MgO$-$SiO_2$ diagram (Figure 12), with negative correlation at 40–50 wt.% MgO and 41–53 wt.% $SiO_2$ (recalculated to dry compositions), while the correlations of $FeO_{tot}$ with $SiO_2$ and MgO are insignificant. The Mg# of peridotite from different bodies ranges within 0.89–0.92, irrespective of orthopyroxene percentage, and corresponds to that in mantle rocks. Most of the samples contain less than 0.5 wt% CaO, which indicates the absence or very low amounts of clinopyroxene and amphibole. The low contents of CaO and $Al_2O_3$ not exceeding 2 wt.% in peridotites demonstrate the absence of lherzolite. The Olkhon peridotite is almost free from serpentine, but the ubiquitous presence of clinochlore (a few percent) accounts for 1%–2% loss on ignition in most of the analyses (Table S5 in Supplementary Materials).

The sum of REE in dunite and harzburgite is within the chondrite values (Table S6 in Supplementary Materials). The most depleted peridotites have the subchondritic REE distribution with $La_n/Yb_n = 1.5$–3.1 and without significant anomalies (Figure 13), which is consistent with earlier published data [23]. However, some samples have fractionated negative REE patterns with $La_n/Yb_n$ up to 7–10 and notable LREE enrichment, though similar HREE with respect to the depleted varieties. These REE patterns may indicate secondary enrichment of the peridotites during a metasomatic event of yet-unknown age and origin. Many rocks show positive Sr and Ta anomalies in multi-element spectra, at a negative or absent Nb anomaly.

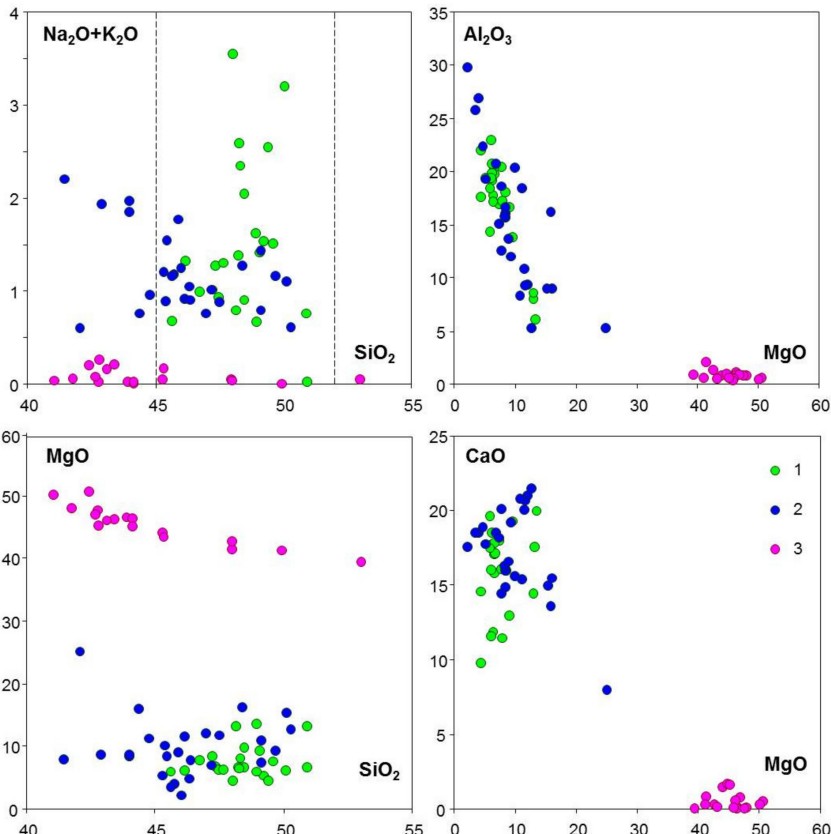

**Figure 12.** Ophiolitic gabbro and peridotites in major-element variation diagrams. 1, 2 = gabbro of 1 and 2 types (see Table S5 in Supplementary Materials); 3 = dunite and harzburgite.

Gabbros have subalkaline compositions according to the contents of silica and alkalis (Figure 12) and belong to two groups according to LREE and MREE distribution: (1) Treugolnik, Gantel, and Krest and (2) Orgoita, Olkhon, and Tankhan. Major-element compositions in the two groups differ only in the field of $Na_2O + K_2O > 1.5$ wt.%, where $SiO_2$ is lower in group 2 than in group 1 (Figure 12), but overlap in the field of low alkalis. The contents of MgO show prominent negative correlation with $Al_2O_3$ but do not correlate with rather high CaO (Figure 12), because low-silica plagioclase and clinopyroxene are main phases, while olivine is subordinate. Unlike the Krestovsky gabbro in the southwestern Olkhon terrane, these bodies in the central and northeastern parts are free from biotite, have low $K_2O$ (<0.4 wt.%), and contain small percentages of opaque phases. Magnetite is accessory in most of the samples, though, being present in leucogabbro, likely as a late crystallization product.

REE abundances in the Olkhon gabbros are no higher than 20 chondritic units (Figure 13). Those of REE group 1 (Treugolnik, Gantel, and Krest) have flat chondrite-normalized REE patterns, $La_n/Yb_n$ = 1 to 2.5 and slightly decreasing abundances from LREE to MREE ($La_n/Sm_n$ = 1.1–1.6). The Orgoita, Olkhon, and Tankhan gabbros have low contents of LREE: $La_n/Sm_n$ = 0.55–0.8 at $La_n/Yb_n$ = 0.7–1.4. The difference in REE patterns suggests that these bodies are fragments of different intrusions that formed by fractionation of chemically different melts. Almost all gabbros have positive Eu/Eu* up to 2.3 due to fractionation of low-silica plagioclase that crystallized at low oxygen fugacity.

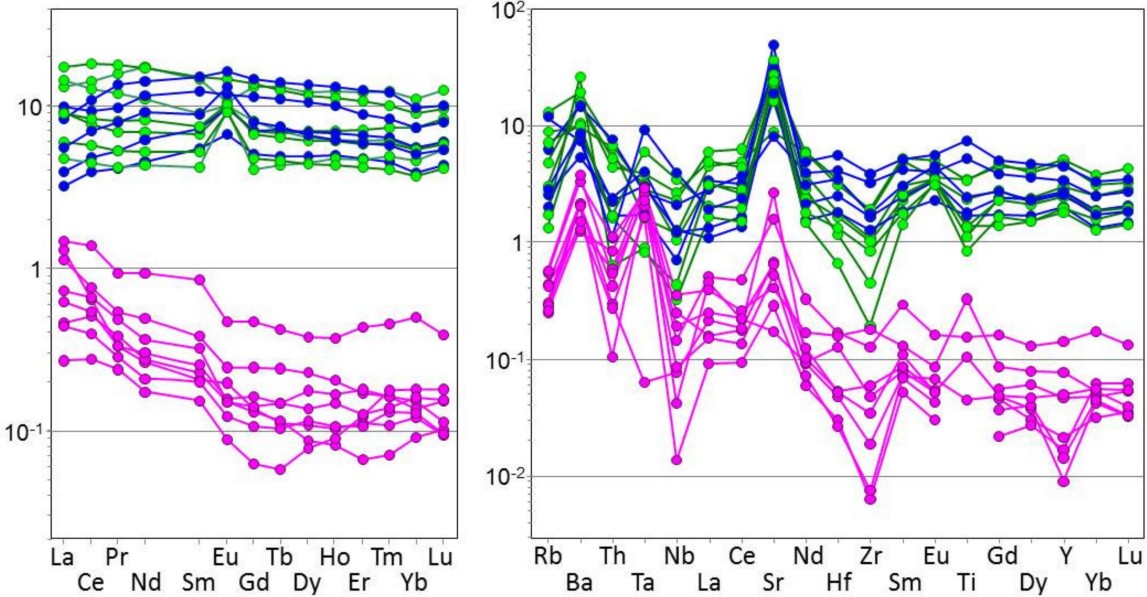

**Figure 13.** Chondrite-normalized rare earth element and primitive mantle-normalized trace element spectra in ophiolitic gabbros and peridotites. Color legend is the same as in Figure 12. Chondrite and primitive mantle compositions are after [59].

The multi-element spectra (Figure 13) have a prominent positive Sr anomaly due to the presence of low-silica plagioclase, as well as an Eu anomaly. The negative Ta-Nb anomaly appears in half of samples but lacks in the other samples, which even have Ta peaks. In the Nb/Yb–Th/Yb diagram (Figure 14), the gabbro compositions plot between MORB and suprasubduction zone basalts and thus may have originated in a back-arc basin setting.

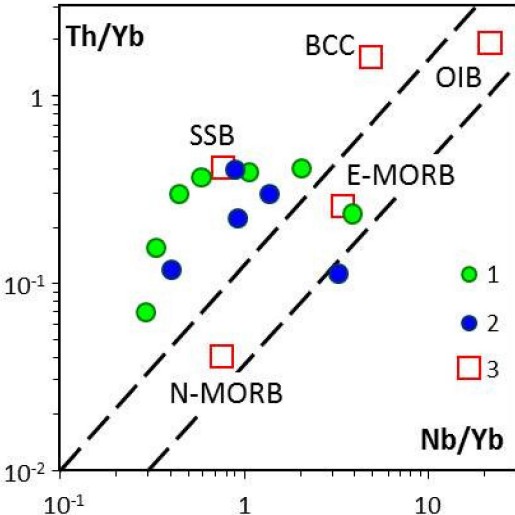

**Figure 14.** Ophiolitic gabbro in the Th/Yb-Nb/Yb diagram. 1 = type 1 gabbro; 2 = type 2 gabbro; 3 = average ratios in basalts of different tectonic settings: N-MORB and E-MORB = mid-ocean ridge [59], OIB = oceanic islands [59], SSB = supra-subduction [60], BCC = bulk continental crust [61].

## 7. Ages of Peridotites

The ages of different rocks of ophiolitic affinity in the Olkhon terrane remain poorly constrained. Numerous attempts of separating zircon or baddeleyite from the gabbros had no success. A SHRIMP-II U-Pb Middle Ordovician age of 467 ± 1.8 Ma became recently available only for zircon from apogranitic plagioclasite that cuts the Shida peridotite [23]. This age may be considered only as an upper limit for the Shida peridotite.

Accessory zircons in the Tog peridotite samples separated from coarse-grained amphibole–chlorite–magnetite rocks (presumably metasomatic) in our study are mostly 10 to 500 µm short-prismatic zircons with rounded edges and corners, or fragments of such crystals. Most of them have a complex patchy texture (Figure 15), with domains looking like relict partly resorbed cores in cathodoluminescence images (grains 17, 19, and 25 in Figure 15), zones of deformation and recrystallization, healed cracks, and other signatures of various effects. Some retain relict domains of rhythmic growth zoning typical of magmatic zircons, which are interrupted by later zircon generations. Generally, the shapes and textures of zircons rather indicate their metamorphic origin.

The zircons mainly have low contents of U (2–17 ppm) and Th (1–20 ppm) common to zircons from ultramafic and mafic rocks, and Th/U ratios from 0.1 to 1.8. Laser ablation U-Pb dating of 36 zircon grains in total (Table S7 in Supplementary Materials) yielded a consistent population with a 465 ± 3 Ma concordant age (mean square weighted deviation (MSWD) = 0.0038), irrespective of grain morphology, structure, and zoning (Figure 16). The Middle Ordovician age of the zircons from the amphibole–chlorite–magnetite rocks almost coincides with the published data for zircons from plagioclasite veins in the Shida peridotite [23] and for amphibolite metamorphism in the Olkhon terrane [20]. This age may be assumed as an upper limit of the Tog peridotite.

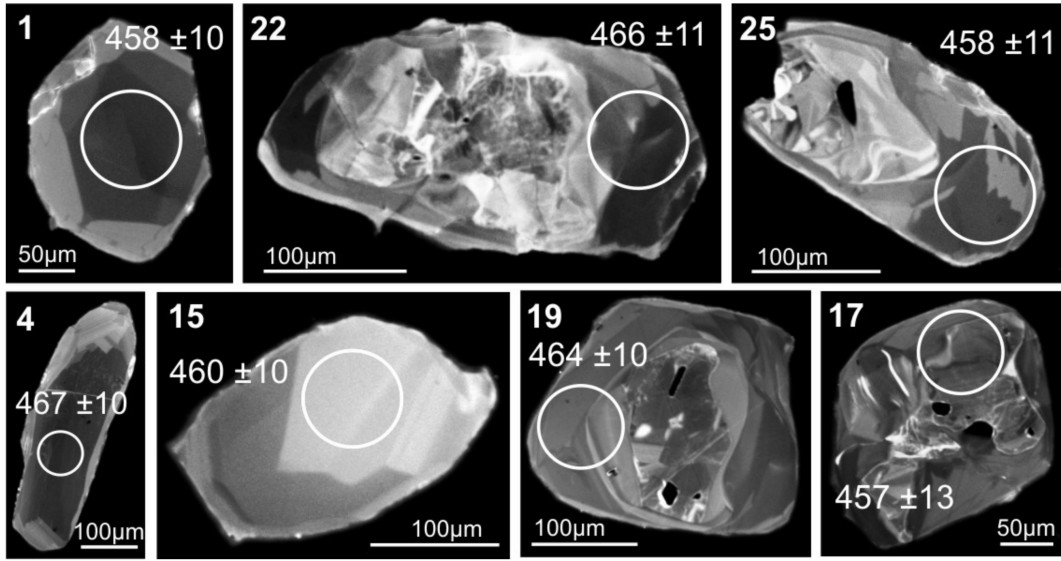

**Figure 15.** Cathodoluminescence (CL) images of representative zircon crystals from coarse-grained amphibole–chlorite–magnetite rocks (metamafic) in the Tog peridotite. Spot location and the respective individual $^{206}$Pb/$^{238}$U ages with 1-sigma uncertainty. Numbers in the left corner correspond to numbers in Table S7 in Supplementary Materials.

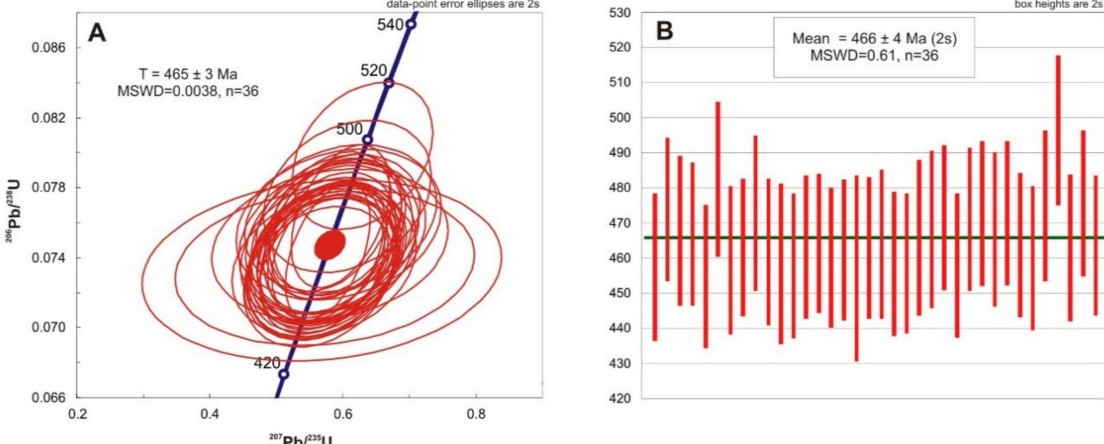

**Figure 16.** (**A**) Concordia (Wetherill) diagram and (**B**) weighted average LA–SF–ICP–MS $^{206}$Pb/$^{238}$U ages of zircon from the Tog peridotite. Error ellipses/bars are at the 2-sima levels. MSWD = mean square weighted deviation.

## 8. Discussion

The diverse gabbroic, amphibolite, and metamorphic mafic rocks from the Olkhon terrane may belong to an ophiolite complex which has a larger occurrence in the terrane than was previously believed. The gabbros appear as small or large tectonic blocks among amphibolites and, in their turn, experienced effects of the respective PT conditions. The almost serpentine-free compositions of the Olkhon peridotites may record rheomorphic origin. According to a previous model [23,39], they might be interpreted as resulting from de-serpentinization during metamorphism. However, we rather suggest that the peridotites affected by amphibolite metamorphism originally contained little or no serpentine and were modified at temperatures above the serpentine stability field [54,62]. This hypothesis is supported by several features of peridotites. First of all, the Tog and Shida peridotites lack forsterite and enstatite with #Mg = 0.96–0.99 which commonly form during de-serpentinization. The Fe-Mg silicates preserve Mg# typical of mantle peridotites, except for slightly higher Mg# values (up to 0.94)

in olivine from the Shida peridotite [23], indicating its equilibrium with chromium-bearing magnetite at the amphibolite facies conditions. The presence of 1–2 wt.% $Al_2O_3$ in orthopyroxene is inconsistent with de-serpentinization, though low CaO (Table S1 in Supplementary Materials) may be evidence of low-temperature origin. The rocks are free from antigorite and talc, which are common phases in peridotites subjected to progressive metamorphism. The olivine–orthopyroxene–Cr-spinel–chlorite assemblage typical of the Shida, Bayar, Sakhyurta, and Tog peridotites is evidence of equilibrium in the amphibolite facies conditions since chlorite becomes unstable at ~700 °C. Cr-spinel equilibrated with chlorite has low contents of Al and Mg.

The pleonast–enstatite varieties found in the Turpan peridotite may result from its high-temperature synmetamorphic metasomatism with inputs of $Al_2O_3$, which is consistent with high Al contents in related amphibolite containing garnet uncommon for the Olkhon mafic metamorphic rocks.

The lithologies and compositions of peridotites correspond to the harzburgite type [63,64] of suprasubduction-zone (SSZ) ophiolites [3,4,65] widespread in the Mediterranean region [14].

The Olkhon gabbros vary from high-Mg melanocratic rocks compositionally close to pyroxenite to leucogabbro and anorthosite. Most of them have relatively high CaO(Figure 10). Ophiolitic gabbros are of two types with low or intermediate $TiO_2$ contents: <0.5 wt.% and >0.5 wt.%, respectively. The former belongs to a cumulative peridotite–gabbro sequence occurring above the petrographic Moxo boundary in ophiolite suites [1]. The latter (so-called isotropic gabbro) is restricted to the upper part of ophiolites and has genetic relations with sheeted dike complexes and volcanic rocks. Most of the Olkhon gabbros are geochemically similar to those from cumulative ophiolites [58,66–70]. Their positive Sr anomaly and negative Ta-Nb anomaly make the Olkhon gabbro similar to SSZ ophiolites [3,4].

According to their structure, the central and northeastern parts of the Olkhon terrane, with numerous small peridotite and gabbro bodies, as well as ophiolitic volcanics and subvolcanics linearized in the course of synmetamorphic strike-slip faulting, correspond to accretionary wedges known from younger subduction-accretionary orogens enclosing ophiolite fragments, e.g., Mesozoic–Cenozoic complexes in the Russain Far East [58,71]. In this respect, it might be reasonable to interpret the central and northeastern Olkhon terrane as a metamorphosed accretionary wedge, which remains little deformed and includes metamorphic blocks that differ in age and composition of protoliths [19], more so that it is adjacent to the Krestovsky island arc subterrane in the southeast. However, there are some considerations against this interpretation. Accretionary wedges form in a fluid-saturated upper crust where primary mineral assemblages of small peridotitic bodies become replaced by hydrous phases, such as serpentine, chlorite, tremolite, talc, and carbonates [72], like that in the Mesozoic Samarka terrane from Sikhote-Alin [57]. However, the Olkhon peridotites were almost free from serpentine before the regional metamorphism and hardly could be involved into the orogenic framework as small bodies at shallow depths.

The previous model [23], implying that peridotites from an underlying ophiolite (in the present erosion surface) would be incorporated into the terrane structure during a collision, appears open to criticism. First, the presence of such an ophiolite in the deep structure, as well as its obduction onto the Siberian craton, has no geophysical or geodynamic support. Second, it is unclear why denser and heavier peridotites with negative buoyancy would intrude into shallower crust as small (meters long) bodies during orogenesis.

A more plausible scenario is that the orogen involved quite a thick ophiolite, with minor low-temperature alteration of primary mineral assemblages, during a frontal collision. As a result of collisional crust thickening, the ophiolite moved down to middle and lower crust and was affected by high-temperature metamorphism of amphibolite facies. This led to regeneration of poorly serpentinized peridotite, re-equilibration of mantle mineral assemblages, and formation of clinochlore. Gabbro also underwent transformations and became partly replaced by scarn associations. When the frontal collision changed to an oblique one, with strike-slip faulting, the rock complexes in the collisional orogen, including the ophiolite, broke up into dispersed small bodies detectable in the present structure.

## 9. Conclusions

1. Numerous dispersed ophiolite fragments occur in the Lower Paleozoic Olkhon terrane (Baikal collisional belt, southern periphery of the Siberian craton) as n–n × 100 m peridotite and up to 1 km gabbro bodies. The peridotites are mostly dunite and harzburgite and less abundant orthopyroxenite, wehrlite, and clinopyroxenite, while the gabbros vary from leuco- to melagabbro according to relative proportions of clinopyroxene, olivine, and plagioclase. Clinopyroxenite and anorthosite occur as thin layers or lenses among the gabbro.

2. The peridotite bodies correspond to highly depleted harzburgite-type associations typical of suprasubduction-zone ophiolites, while the geochemical signatures of gabbro provide evidence of their origin in a back-arc basin setting which existed in the southern craton margin in the Neoproterozoic [73]. The gabbros are of two geochemical types that represent, respectively, cumulative gabbro–peridotite sequences and upper isotropic gabbro of ophiolite complexes. Most of them show Ta-Nb minimums and Sr maximums in spider diagrams, as in suprasubduction ophiolitic gabbro [3,4,64].

3. Mineral assemblages in the Olkhon peridotites (Ol + Opx + Chl + Chr) may result from amphibolite regional metamorphism at 500–600 °C. The related high-temperature metasomatism produced olivine–pleonast–enstatite and anorthite–garnet–fassaite mineral assemblages and garnet amphibolites.

4. The obtained 465 ± 3 Ma age of zircon from the metasomatic magnetite–amphibole–chlorite rock in the Tog peridotite is coeval with that for plagioclasite veins in the Shida peridotite [23] and corresponds to the time of amphibolite regional metamorphism in the Olkhon terrane. The ophiolite age is unknown but may be Neoproterozoic to a high probability.

5. Although the occurrence of peridotite and gabbro bodies corresponds to a setting of an accretionary wedge metamorphosed during collisional orogenic processes, there is evidence that a large ophiolite slab was incorporated into the orogen during the frontal collision of the Olkhon composite terrane with the Siberian craton and was broken up into small dispersed bodies during strike-slip deformations in a later oblique collision event.

**Supplementary Materials:** The following are available online at http://www.mdpi.com/2075-163X/10/4/305/s1, Table S1: Mineral chemistry of silicates and ilmenite from peridotites [74], Table S2: Chemistry of spinel-group minerals from peridotites [74], Table S3: Mineral chemistry of pyroxenes from anorthite-fassaite-garnet and anorthite-amphibole-garnet rocks associated with peridotites, Table S4: Mineral chemistry of garnet from anorthite-fassaite-garnet and anorthite-amphibole-garnet rocks associated with peridotites, Table S5: Whole-rock major oxide (wt%) and some trace element (ppm) in mafic and ultramafic rocks of Olkhon Terrain, Table S6: Whole-rock trace and RE element (ppm) in mafic and ultramafic rocks of Olkhon Terrain, Table S7: Zircon data for the Tog amphibole-magnetite-chlorite rocks.

**Author Contributions:** E.V.S. did geological study, collected samples, interpreted the data, prepared tables and figures, and wrote the manuscript; V.S.F. did geological study; A.V.L. did geological study, collected samples, prepared figures and tables, and wrote the manuscript; E.V.P. interpreted the data, prepared figures, and wrote the manuscript; D.V.S. performed analytical study, prepared tables and figures, and wrote the manuscript; A.E.S. did geological study, collected samples, and prepared figures. All authors have read and agreed to the published version of the manuscript.

**Funding:** The study was carried out on government assignments to different institutions: Institute of the Earth's Crust (Irkutsk), Institute of Geology and Mineralogy (Novosibirsk), Institute of Geology and Geochemistry (Ekaterinburg), and Geological Institute (Moscow). Geological investigation of peridotites and gabbros was supported by the Russian Foundation for Basic Research (Project 20-05-00005). Mineral compositions were analyzed under the support of the Russian Science Foundation (Project 18-17-00101), and dating of zircons was funded by the government of the Russian Federation (Project 075-15-2019-1883).

**Acknowledgments:** Thoughtful comments by three anonymous reviewers on this manuscript are gratefully acknowledged. We wish to thank T. Perepelova and O. Sklyarova for assistance in manuscript preparation.

**Conflicts of Interest:** The authors declare no conflict of interest.

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
