# Peer review of "Dismembered Ophiolite of the Olkhon Composite Terrane (Baikal, Russia): Petrology and Emplacement"

_minerals, doi:10.3390/min10040305_

Round 1

Reviewer 1 Report

This paper presents detail field work, petrology, mineralogy, geochemistry and U-Pb dating of new founded ophiolite in Olkhon Terrane. The discussion of field relationship, tectonic setting, geochmical petrogenisis and the significance of U-Pb ages are reasonable. This could be contribution to realize the tectonic evolution of Central Asian Orogenic Belt.

I recommend publication in Minerals after minor revise which include some questions should declare by the authors, the figures modify and many misprints, which I have marked in the PDF.

Author Response

Thank you very much! Please see attached file.

Reviewer 2 Report

Review of the manuscript entitled “Dismembered Ophiolite of the Olkhon Composite Terrane (Baikal, Russia): Petrology and Emplacement” (authors Sklyarov et al.)

The paper describes very detailed field observation, petrography and geochemistry data a complex, a tectonically emplaced petrographic association representing dismembered ophiolites in the Early Paleozoic Olkhon terrane. Based on the depleted character of peridotites, the authors argue for a Super Subduction Zone (SSZ) interpretation. The associated basic rocks bear a back-arc geochemical character. Two group signatures were recognized in the basic rocks, both groups being interpreted as SSZ, based on low Ta/Nb ratio and positive Sr anomalies normalized to chondrite. Based on the metamorphic assemblages described in the ultramafic rocks the authors consider that peridotites underwent low amphibolite and amphibolite regional metamorphism at 500-650°C, suggesting the formation in an accretionary wedge metamorphosed during a frontal collisional orogeny, formed a part of the peri-oceanic structures of the Siberian craton. Later oblique collision event dismembered the rock suite by strike-slip faulting.

The text is very well written. The abundance of field and petrographic details gives confidence in detailed and careful scientific work. The photographs, photomicrographs, and plots are of very good quality and the overall scientific content and soundness are excellent. It was a pleasure to review this manuscript.

This paper is an excellent and very useful contribution to integration the understanding of the Paleozoic Olkhon terrane in the framework of southern peri-Siberian craton evolution. However, minor adjustments, changes, and clarifications are needed (see detailed comments annotated on the pdf document attached).

I recommend the publication in Minerals after minor modifications.

General comments

  1. Terminology

Names used for certain minerals are wrong, or some are obsolete. The nomenclature of spinel-group minerals and clinopyroxene should be revisited. For example, ferrichromite or aluminochromite are not accepted terms. If Al > Cr (apfu) and Fe2+ > Mg, the mineral is hercynite. If Mg > Fe2+ and Cr > Al the mineral is magnesiochromite. A chromium-rich hercynite is simply a chromian hercynite. If Fe3+ > Al (apfu) the mineral is magnetite. Please check the IMA mineral list for accepted end-members and mineral names (https://rruff-2.geo.arizona.edu/ima/#)

The fassaite name for clinopyroxene is no longer accepted by IMA. As it was first described in 1920s, it is a diopside with significant Al and Fe3+. In the 1990s other people redefined it as low Fe augite, creating even more confusion. Now, this type of composition is called simply augite. More than that, your table 3 (SupplTables) shows Fe2+ >> F3+, so not even after this criterion we cannot call this mineral fassaite, as it is misleading, not only obsolete. High Al and Ca in augite simply shows high Ca-Tsch molecule in cpx, suggesting high T crystallization.

High-Mg chlorite should be named as a mineral phase: clinochlore

  1. Few comments that can be addressed by clarifications in the text:
  • The interpretation as SSZ based mostly on the depleted character of the peridotites is not always safe. All mantle harzburgites in the mantle are depleted. The same applies to the basic rocks. The low Ta/Nb in some samples and high Sr simply suggest the cumulus character of plagioclase correlated to the high amount of oxides that accommodate Nb (high Nb values associate with high Ti values). Associated high values of Th/Yb and Nb/Yb might suggest prior fractionation of Yb in garnet crystallization in the source melting of the basaltic rocks. This is not a character characteristic for SSZ, only, it is a much more general situation encountered in other geotectonic settings of basic rocks. Are there other criteria (relict mineral assemblages, HP assemblages prior to granulitic or amphibolitic overprint during the collision, others?) to suggest SSZ?
  • It is not clear how Al-rich metasomatism can produce olivine and enstatite, or even garnet. Metasomatism requires a subsolidus transformation of older assemblage into a new assemblage, using the space of the pre-existing assemblage. I agree that they are produced by mineral reactions in the presence of a fluid. However, Al is one of the most immobile elements during metamorphism. The way it is phrased suggests that the Al2O3 was brought into the system by a fluid agent ("aluminous metasomatism"). This can happen if the fluid is, in fact, an Al-rich melt. However, the reaction of such melt with prior phases can show in several types of reaction-related microtextures (mostly symplectites involving plagioclase and/or spinel). This is probably not the case here, or at least nothing it is convincing here about an Al-metasomatism. High-temperature reactions in the "skarn-type" (calc-silicates) environment can easily form the mineral assemblage that you mention. They are metasomatic reactions and they need an open system. However, "aluminous metasomatism" does not sound correct. Please re-evaluate the interpretation, if possible, or provide more clarification.

For detailed comments (notes), see attached pdf.

Good luck!

Reviewer 3 Report

All comments and proposals for correction are noted in the attached manuscript.
